**Brief Communication**

# Segmentation metric misinterpretations in bioimage analysis

Dominik Hirling [1,2], Ervin Tasnadi[1,2], Juan Caicedo[3], Maria V. Caroprese[4], Rickard Sjögren[5,6], Marc Aubreville [7], Krisztian Koos [1] & Peter Horvath [1,8,9] ✉

Quantitative evaluation of image segmentation algorithms is crucial in the field of bioimage analysis. The most common assessment scores, however, are often misinterpreted and multiple definitions coexist with the same name. Here we present the ambiguities of evaluation metrics for segmentation algorithms and show how these misinterpretations can alter leaderboards of influential competitions. We also propose guidelines for how the currently existing problems could be tackled.

In today's scientific environment, with an increasing attention on artificial intelligence solutions for imaging problems, a plethora of new image segmentation and object detection methods have emerged. Thus, quantitative evaluation is crucial for an objective assessment of algorithms. Often, object detection and segmentation tasks use evaluation metrics with the same name but a different meaning due to the differences between object- and pixel-level classification, or just because multiple interpretations coexist. One could argue that, in most cases, the meaning should be clear from the context, however, specific and often nondetailed characteristics of the circumstances (for example, small variations of the task) can make it hard for readers to understand the exact meaning of different metrics. Recently, an exhaustive study has been published on the variety of assessment scores and their proper use-cases[1]. Our study focuses on the various interpretations that have emerged in the research communities related to some segmentation scores. As such, we identified five different definitions for the 'average precision' (AP) and six different interpretations for the 'mean average precision' (mAP) metrics in the literature. To make things even more complicated, even when some methods work with the same dataset, the metrics used for the evaluation of performance are not necessarily the same. The aims of our study are to shed light on some of the main issues with the current state of segmentation and object detection metrics and to investigate the reasons for the ambiguous use of classification concepts. We also point out the problems of using similar metrics with nuanced differences by evaluating the 2018 Kaggle Data Science Bowl (DSB), 2021 Kaggle Sartorius Cell Instance Segmentation and 2021 MIDOG (Mitosis Domain Generalization) challenge submissions with metrics of similar meaning but slightly differing interpretations.

Our study mainly focuses on segmentation scores that are object-based, that is a single object is counted as true positive (TP), false positive (FP) or false negative (FN) instead of labeling every pixel. However, some of the object-based metrics can be defined on a pixel level as well[2–5] (Supplementary Table 1).

Object-based segmentation scores are used (1) for object detection tasks (usually, when we want to find objects in an image with bounding boxes) or (2) for segmentation tasks, when the image contains many objects or when the pixel perfect delineation of the boundaries is not the only and most important priority to be evaluated. When using these scores, as a first step, an intersection over union (IoU) threshold is specified. Any prediction that yields an IoU score greater than this threshold will be considered as true positive, otherwise, we consider it as false positive. If an object cannot be detected with the specified IoU threshold, we consider a false negative label. All the metrics noted above have their own purpose in evaluation pipelines. For example, if one wants to quantify whether every object has been detected, but does not care about false detections, they will prioritize what is known as the recall metric. This metric is also known as sensitivity and is especially important in medical applications. If one cares about quantifying objects and about false detections, $F1$ and threat scores should be prioritized, as they penalize false detections. These scores are more appropriate for cell counting applications. Besides the most important simple metrics listed in Supplementary Table 1, three other important quality assessment scores are often used in image processing tasks:

- The panoptic quality (PQ) metric measures segmentation quality and recognition quality simultaneously:

[1]Biological Research Centre, Eötvös Loránd Research Network (ELKH), Szeged, Hungary. [2]Doctoral School of Computer Science, University of Szeged, Szeged, Hungary. [3]Broad Institute of Harvard and MIT, Cambridge, MA, USA. [4]Sartorius, Corporate Research, Royston, UK. [5]Sartorius, Corporate Research, Umeå, Sweden. [6]CellVoyant Technologies Ltd, Bristol, UK. [7]Technische Hochschule Ingolstadt, Ingolstadt, Germany. [8]Single-Cell Technologies Ltd, Szeged, Hungary. [9]Institute for Molecular Medicine Finland (FIMM), University of Helsinki, Helsinki, Finland. ✉e-mail: horvath.peter@brc.hu

**Table 1 | Various interpretations of AP and mAP**

| Metric | Note | Referenced in |
|---|---|---|
| $AP_1 = \int p(r)\mathrm{d}r$ | Fixed IoU threshold | 10–14 |
| $mAP_1 = \frac{1}{N} \sum_{n=1}^{N} AP_1(n)$ | $AP_1(n)$ is $AP_1$ calculated for class $n$ | 10 |
| $AP_2 = \frac{TP}{TP+FP+FN}$ | Fixed IoU threshold | 15–17 |
| $mAP_2 = \frac{1}{\|\text{thresholds}\|} \sum_{t\in\text{thresholds}} AP_2^{@(IoU=t)}$ | | 18 |
| $AP_3 = \frac{1}{\|\text{thresholds}\|} \sum_{t\in\text{thresholds}} \left(\frac{TP}{TP+FP}\right)^{@(IoU=t)}$ | | 19 |
| $mAP_3 = \frac{1}{\|\text{images}\|} \sum_{i\in\text{images}} AP_3(i)$ | $AP_3(i)$ is calculated for image $i$ | 19 |
| $mAP_4 = p \times r = \frac{TP^2}{TP^2+TP\times FN+TP\times FP+FP\times FN}$ | Also known as 'Digits score' | 20 |
| $mAP_5 = \frac{1}{\|\text{images}\|} \sum_{i\in\text{images}} mAP_2(i)$ | $mAP_2(i)$ is calculated on image $i$ | 21 |
| $AP_4 = \frac{1}{\|\text{thresholds}\|} \sum_{t\in\text{thresholds}} AP_1^{@(IoU=t)}$ | Also known as the 'COCO metric' | 22,23 |
| $mAP_6 = AP_5 = \frac{1}{N} \sum_{n=1}^{N} AP_4(n)$ | $AP_4(n)$ is $AP_4$ calculated for class $n$: also known as the 'primary COCO competition metric' | 23 |

$$PQ = \frac{\sum_{(p,g)\in TP} IoU(p,g)}{TP + \frac{FP+FN}{2}},$$

where $p$ is a predicted object labeled as true positive and $g$ is the corresponding ground truth.

- The AP metric is calculated by taking the area under the precision–recall curve, which is given for recalls $r$ and associated precisions $p(r)$:

$$AP_1 = \int p(r)\mathrm{d}r.$$

- When it comes to multi-class classification or detection, the mAP is used, which is calculated as the average of AP values taken for every individual class:

$$mAP_1 = \frac{1}{N}\sum_{n=1}^{N} AP_1(n).$$

Most of the time, biological image segmentation tasks use evaluation metrics such as the ones defined in Supplementary Table 1 rather than $AP_1$ (which is very common in computer vision) because $AP_1$ requires a confidence value for each detected and/or segmented object and for segmentation, but very few algorithms have such a score.

Despite clear definitions existing for the AP and mAP metrics, many alternative interpretations have emerged recently. This confusion possibly stems from the evaluation section of the 2018 Kaggle Data Science Bowl challenge, where the threat score metric was referred to as 'an AP value' (as seen on the event's webpage), even though this expression is ignored in the related article[6]. Besides this new definition, possible variations of the metrics AP and mAP also started to emerge since then: we have identified six different interpretations for the AP and five different interpretations for the mAP metrics (Table 1).

These variations exist because there is no consensus for what 'mean' and 'average' stand for: for some metrics, 'average' in AP is equivalent to the threat score for a single image, whereas in other cases it stands for the average threat score across several IoU thresholds. As for the mAP metric, 'mean' is sometimes used for the IoU thresholds, whereas in other cases it indicates the average across all images in a given dataset, but there is also precedent for averaging across both IoU thresholds and images. Furthermore, when a metric uses several IoU thresholds, the starting and ending threshold values should be specified along with the threshold step size. This may also contribute to the increasing diversity among the existing evaluation metrics (Fig. 1a–c).

To demonstrate the possible consequences of misinterpreting the metrics, we used several assessment scores to evaluate submissions to prestigious bioimage competitions. We evaluated the second stage submissions of the 2018 Kaggle Data Science Bowl[6], final stage submissions of the 2021 Sartorius[7] and final stage submissions of the 2021 MIDOG[8] challenges (sample images from the competitions can be seen in Extended Data Fig. 1). For evaluation, we used several misinterpretations of AP, mAP and the $F1$ score (as used in ref. 9) with various thresholds, both in an aggregated and in an averaged-by-image way. Our results indicate that the leaderboards of the competitions are substantially influenced by changing the evaluation metric, depending on which properties we modify. On using a fixed IoU threshold, the threshold value can noticeably influence the outcome. However, when using multiple IoU threshold values, tweaking the step size does not change the outcome drastically. Using a fixed IoU threshold of 0.9 yielded generally low correlation values, thus we argue that such a high threshold is, in general, not useful to determine the efficiency of a segmentation algorithm. When we calculate the scores in an image-by-image way and take the average of these values, the outcome is notably different compared to what we get when we calculate the scores in an aggregated way (Fig. 1d–g). Deciding when to use averaging or aggregation mainly comes down to a few aspects: for datasets that either have a high variance in the number of objects in each image or the images are similar to each other (for example, same modalities), the aggregation strategy can be useful, however, for datasets with a consistent number of objects and diverse images (for example, different modalities), the averaging strategy is better due to every image getting the same weight in the final score. We note that despite some metric variations having a high correlation to each other, this does not mean that the scores cannot be used to alter leaderboards. Even a correlation as high as 0.99 may

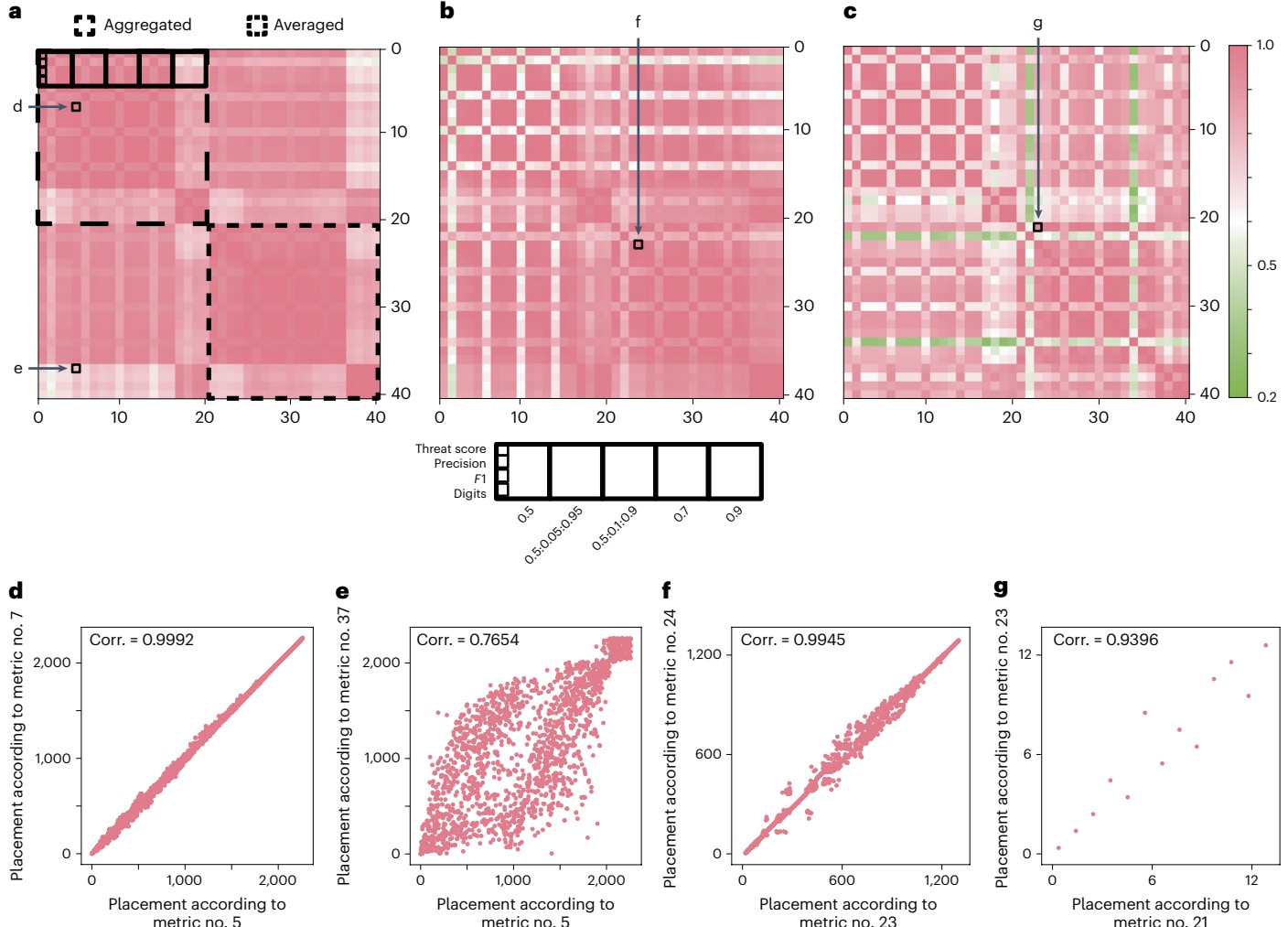

**Fig. 1 | Results for the evaluation of the final stage submissions of three different competitions. a–c**, Cross-correlation matrix of the different metrics for the 2018 Kaggle Data Science Bowl (**a**), 2021 Sartorius Cell Segmentation Challenge (**b**) and the 2021 MIDOG (**c**) challenges. Both image-by-image and aggregated scores are calculated with various IoU threshold ranges for four different metrics. **d**, Correlation (Corr.) between the aggregated version of the threat score with IoU = 0.5:0.05:0.95 and aggregated $F1$ with IoU = 0.5:0.05:0.95 for the 2018 DSB competition. **e**, Correlation between the aggregated version of the threat score with IoU = 0.5:0.05:0.95 and averaged threat score with IoU = 0.9 for the 2018 DSB competition. **f**, Correlation between the averaged version of the $F1$ score with IoU = 0.5 and averaged digits score with IoU = 0.5 for the 2021 Sartorius competition. **g**, Correlation between the averaged version of the threat score with IoU = 0.5 and averaged $F1$ score with IoU = 0.5 for the 2021 MIDOG challenge.

induce that some solutions shift multiple positions back and forth on the leaderboard. These findings confirm that using clearly defined metrics with as few modifications as possible is vital for the transparency of the biomedical image segmentation field. Therefore, here we propose some recommendations that should be followed on defining the evaluation metric for a segmentation task, to avoid confusion as much as possible.

To resolve some of the main issues mentioned above, we would like to propose some concrete measures to increase transparency and clarity when it comes to selecting and defining an assessment metric for semantic segmentation:

(1) The designation of 'AP' and 'mAP' should be omitted when they are not used according to their original definitions. Instead, we propose that the image processing communities should use the designation of 'threat score'.

(2) IoU threshold ranges and step sizes should be denoted explicitly.

(3) Whether a metric is aggregated through an entire dataset or averaged over the images in the set should also be clearly visible and should be noted explicitly.

(4) Even though it may be implied by the context, it should be stated explicitly whether a metric is calculated pixel-wise or object-wise.

For example, when we would like to use the aggregated threat score from IoU threshold 0.5 to 0.95 with a step size of 0.05, the following notation should be used:

$$TS_{agg}^{0.5:0.05:0.95},$$

whereas in case of an $F1$ score calculated for an IoU threshold of 0.5 averaged over the images in the test set, the notation should be the following:

$$F1_{avg}^{0.5}.$$

Besides these recommendations, we would also advise the researchers to use well-established metrics on datasets on which evaluations have already been executed previously. Whenever evaluated, the same performance scores should be applied on these types of datasets,

or, in a case where a new metric is introduced, the reasons for its introduction should be clearly justified. Once a different metric is used, the 'original' score should also be computed and shown.

To summarize, we have presented some of the main issues related to the variations of evaluation metrics in image segmentation. The possible exploitations and faulty uses have been demonstrated by evaluating prestigious bioimage segmentation challenge submissions via different metric interpretations. We are concerned that a simple approach of consistency, including explicitly specifying the IoU threshold ranges, the averaging strategy and whether the metric is calculated pixel-wise or object-wise, would help to avoid most of the ambiguity related to segmentation tasks in the future. We hope that these recommendations will be of use for the research community.

## Online content

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

## Methods

### Competitions

**2018 Kaggle Data Science Bowl (DSB2018).** The task of this competition was to segment nuclei on microscopy images from approximately 15 different biological experiments donated by multiple laboratories (a total of 106 images). The images can be divided into two main categories: fluorescent and brightfield microscopy images. A total of 2,263 submissions were processed in the second stage evaluation.

**2021 Sartorius Cell Instance Segmentation Challenge (Sartorius).** Here the task was to segment neuronal cells in light microscopy images, specifically with the SH-SY5Y cell line due to the challenging nature of the task (concave shapes, irregular morphology of cells). A total of 1,304 submissions were processed in the last stage evaluation.

**2021 Mitosis Domain Generalization Challenge (MIDOG2021).** A challenge that focused on detection rather than segmentation, this competition included brightfield hematoxylin and eosin-stained microscopy images from six different scanners, with samples from 300 patients with breast cancer. The task was to find the mitotic cells in these images. The final evaluation stage included 13 submissions.

### Ranking

To see how various metric interpretations can alter the leaderboard of competitions, we first processed the ground truth and submission files we received from the challenge organizers: for the 2018 DSB and 2021 Sartorius challenges, we got the run-length encoded representation of the segmentations. As for the MIDOG challenge, we got the centroids of the bounding boxes for every mitotic cell in JSON format.

First, we created labeled mask images from all the data that we received. After that, we used the scripts provided by StarDist (https://github.com/stardist/stardist) as a basis for our evaluation, in which we calculated the submission score for every team according to various metric interpretations. In the end, we calculated the correlation coefficient (Pearson product-moment correlation coefficients calculated with the numpy python library) of the matrix containing every submission score in the challenges (matrix rows, submission of one team according to various metrics).

We note that the $AP_2$ and $F1$ scores are deterministically related, thus, when using a fixed IoU threshold, the correlation between the two is 1. This relation, however, fades away when changing the metric parameters (averaging across multiple IoU thresholds or calculating one metric aggregated, the other one in an averaged-by-image way).

### Reporting summary

Further information on research design is available in the Nature Portfolio Reporting Summary linked to this article.

### Data availability

Images and corresponding ground truth masks are publicly available for the DSB2018 and Sartorius Challenges. As for the MIDOG2021 challenge, the images for the final stage evaluation data are private and thus not available. Submission files from the competitors are available upon request. DSB2018 data https://www.kaggle.com/competitions/data-science-bowl-2018/data. Sartorius data: https://www.kaggle.com/competitions/sartorius-cell-instance-segmentation/data. MIDOG2021 data: https://imig.science/midog2021/download-dataset/.

### Code availability

All of the source code used to process the submissions and create the ranking correlations can be found at https://bitbucket.org/biomag/metric-code/

### Acknowledgements

D.H., K.K., E.T. and P.H. acknowledge support from the Lendület BIOMAG grant (no. 2018–342), TKP2021-EGA09, H2020-COMPASS-ERAPerMed, CZI Deep Visual Proteomics, H2020-DiscovAIR, H2020-Fair-CHARM, HAS-NAP3, Horizon Europe BIALYMP, the ELKH-Excellence grant from OTKA-SNN no. 139455/ARRS, the FIMM High Content Imaging and Analysis Unit (FIMM-HCA; HiLIFE-HELMI), and Finnish Cancer Society. D.H. and P.H. acknowledge the professional support of the Doctoral Student Scholarship Program of the Co-operative Doctoral Program of the Ministry of Innovation and Technology financed from the National Research, Development and Innovation Fund. We acknowledge support from A. Carpenter for the help in sharing the DSB2018 dataset.

### Author contributions

D.H. was responsible for conceptualization, methodology, software, validation, formal analysis, investigation, writing the original draft, review, editing and visualization. E.T. was responsible for software and validation. J.C. helped to write the original draft and helped with review and editing. M.V.C., R.S. and M.A. helped to write the original draft, and with the review and editing. K.K. worked on conceptualization, writing the original draft, review and editing and supervision. P.H. worked on conceptualization, writing the original draft, review and editing, supervision, project administration and funding acquisition.

### Competing interests

J.C. was one of the organizers of the 2018 Kaggle Data Science Bowl. M.V.C. is an employee of Sartorius. R.S. was one of the organizers of the 2021 Kaggle Sartorius Cell Instance Segmentation challenge. M.A. is one of the organizers of the MIDOG challenges. The other authors declare no competing interests.

### Additional information

**Extended data** is available for this paper at https://doi.org/10.1038/s41592-023-01942-8.

**Correspondence and requests for materials** should be addressed to Peter Horvath.

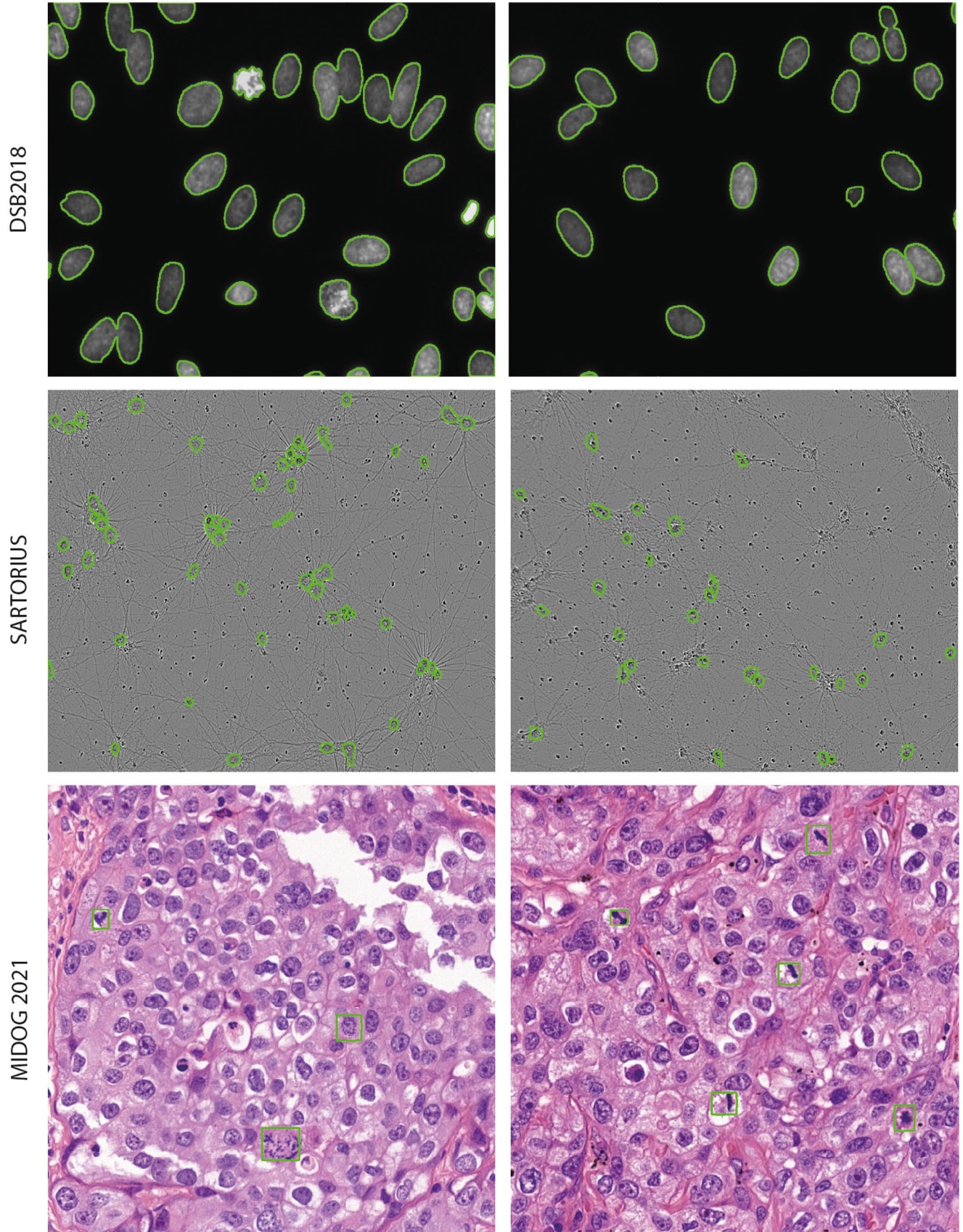

**Extended Data Fig. 1 | Sample images and ground truth labels (in green) from the DSB2018, Sartorius and MIDOG 2021 challenges.**

# Reporting Summary

## Statistics

For all statistical analyses, confirm that the following items are present in the figure legend, table legend, main text, or Methods section.

| n/a | Confirmed | |
|---|---|---|
| ☐ | ☒ | The exact sample size (*n*) for each experimental group/condition, given as a discrete number and unit of measurement |
| ☒ | ☐ | A statement on whether measurements were taken from distinct samples or whether the same sample was measured repeatedly |
| ☒ | ☐ | The statistical test(s) used AND whether they are one- or two-sided *Only common tests should be described solely by name; describe more complex techniques in the Methods section.* |
| ☒ | ☐ | A description of all covariates tested |
| ☒ | ☐ | A description of any assumptions or corrections, such as tests of normality and adjustment for multiple comparisons |
| ☐ | ☒ | A full description of the statistical parameters including central tendency (e.g. means) or other basic estimates (e.g. regression coefficient) AND variation (e.g. standard deviation) or associated estimates of uncertainty (e.g. confidence intervals) |
| ☒ | ☐ | For null hypothesis testing, the test statistic (e.g. *F*, *t*, *r*) with confidence intervals, effect sizes, degrees of freedom and *P* value noted *Give P values as exact values whenever suitable.* |
| ☒ | ☐ | For Bayesian analysis, information on the choice of priors and Markov chain Monte Carlo settings |
| ☒ | ☐ | For hierarchical and complex designs, identification of the appropriate level for tests and full reporting of outcomes |
| ☒ | ☐ | Estimates of effect sizes (e.g. Cohen's *d*, Pearson's *r*), indicating how they were calculated |

*Our web collection on statistics for biologists contains articles on many of the points above.*

## Software and code

Policy information about availability of computer code

| Data collection | No software was used for data collection. |
|---|---|
| Data analysis | We used some of the evaluation code provided by StarDist (https://github.com/stardist/stardist, version 0.8.3). All of the source code used to process the submissions and create the ranking correlations can be found at https://bitbucket.org/biomag/metric-code/ |

For manuscripts utilizing custom algorithms or software that are central to the research but not yet described in published literature, software must be made available to editors and reviewers. We strongly encourage code deposition in a community repository (e.g. GitHub). See the Nature Portfolio guidelines for submitting code & software for further information.

## Data

Policy information about availability of data

All manuscripts must include a data availability statement. This statement should provide the following information, where applicable:
- Accession codes, unique identifiers, or web links for publicly available datasets
- A description of any restrictions on data availability
- For clinical datasets or third party data, please ensure that the statement adheres to our policy

Images and corresponding ground truth masks are publicly available for the DSB2018 and Sartorius Challenges. As for the MIDOG2021 challenge, the final stage evaluation data is private and thus not available. Submission files from the competitors are available upon request.
DSB2018 data: https://www.kaggle.com/competitions/data-science-bowl-2018/data

## Human research participants

Policy information about studies involving human research participants and Sex and Gender in Research.

| | |
|---|---|
| Reporting on sex and gender | Our study did not involve human research participants. |
| Population characteristics | Our study did not involve human research participants. |
| Recruitment | Our study did not involve human research participants. |
| Ethics oversight | Our study did not involve human research participants. |

Note that full information on the approval of the study protocol must also be provided in the manuscript.

# Field-specific reporting

Please select the one below that is the best fit for your research. If you are not sure, read the appropriate sections before making your selection.

☒ Life sciences          ☐ Behavioural & social sciences          ☐ Ecological, evolutionary & environmental sciences

For a reference copy of the document with all sections, see nature.com/documents/nr-reporting-summary-flat.pdf

# Life sciences study design

All studies must disclose on these points even when the disclosure is negative.

| | |
|---|---|
| Sample size | We provide sample sizes from 3 different competitions:<br>The DSB2018 competition included a total of 2263 submissions for the second stage evaluation,<br>the Sartorius challenge included 1304 submissions for the final stage evaluation,<br>the MIDOG2021 challenge included 13 submissions for the final stage evaluation.<br>We included every submission available from the competitions for our analyses, thus, these sample sizes were sufficient for the experiments. |
| Data exclusions | No data was excluded from the analyses. |
| Replication | Running the evaluation scripts on the submission data will reproduce the data of the study. |
| Randomization | Not relevant to study, because we included all of the submissions from 3 different competitions, calculating the correlation between the participants' placements according to different metrics does not require any randomization of the samples/submissions. |
| Blinding | No group allocation was done, thus blinding was also not relevant to study. |

# Reporting for specific materials, systems and methods

We require information from authors about some types of materials, experimental systems and methods used in many studies. Here, indicate whether each material, system or method listed is relevant to your study. If you are not sure if a list item applies to your research, read the appropriate section before selecting a response.

### Materials & experimental systems

| n/a | Involved in the study |
|---|---|
| ☒ ☐ | Antibodies |
| ☒ ☐ | Eukaryotic cell lines |
| ☒ ☐ | Palaeontology and archaeology |
| ☒ ☐ | Animals and other organisms |
| ☒ ☐ | Clinical data |
| ☒ ☐ | Dual use research of concern |

### Methods

| n/a | Involved in the study |
|---|---|
| ☒ ☐ | ChIP-seq |
| ☒ ☐ | Flow cytometry |
| ☒ ☐ | MRI-based neuroimaging |

