## [Peer Review File · Nature Methods]

Peer Review Information

Manuscript Title: Segmentation Metric Misinterpretations in Bioimage Analysis

Corresponding author name(s): Péter Horváth

Editorial Notes: n/a

Reviewer Comments & Decisions:

Decision Letter, initial version:

Dear Peter,

Your Comment, "Segmentation Metric Misinterpretations in Bioimage Analysis", has now been seen by two reviewers. As you will see from their comments below, the reviewers find your Comment of considerable potential interest but have a number of concerns. We are interested in the possibility of publishing your paper in Nature Methods, but would like to consider your response to these concerns before we reach a final decision on publication.

We ask that in response to the referee comments, you focus your revisions on (1) making the piece more accessible to a broad audience, (2) better explaining not just what metrics are being used but why (see points made by ref 2), and (3) describing how the points made are broadly valid despite your use case being focused on one competition. We appreciate that this may add length to the piece.

In order to facilitate review of the revised manuscript please underline any additions to the text or areas with other significant changes. Please ensure that the revised version is as concise as possible, and that it conforms to our format requirements (see <http://www.nature.com/nmeth> for our Guide to Authors). We shall return to the issue of format and presentation in the event that the manuscript is accepted for publication.

Nature Methods is committed to improving transparency in authorship. As part of our efforts in this direction, we are now requesting that all authors identified as 'corresponding author' on published papers create and link their Open Researcher and Contributor Identifier (ORCID) with their account on the Manuscript Tracking System (MTS), prior to acceptance. This applies to primary research papers only. ORCID helps the scientific community achieve unambiguous attribution of all scholarly

contributions. You can create and link your ORCID from the home page of the MTS by clicking on 'Modify my Springer Nature account'. For more information please visit www.springernature.com/orcid.

We shall hope to receive your revised version as soon as possible. If you anticipate a considerable delay please let us know.

Please use the link below when you are prepared to resubmit.

[Redacted]

The above URL links to your confidential home page and associated information about manuscripts you may have submitted, or that you are reviewing for us. If you wish to forward this email to co-authors, please delete the link to your homepage.

Sincerely,
Rita

Rita Strack, Ph.D.
Senior Editor
Nature Methods

Reviewers' Comments:

Reviewer #1:

Remarks to the Author:

The manuscript entitled 'Segmentation metric misinterpretations in bioimage analysis' aims to shed light on some of the issues that appear due to misinterpretation of the definition of the segmentation scores in biomedical images. After a brief introduction and the definition of the main metrics. The manuscript uses the extensively used Average Precision (AP) metric and its variations as a use case. It comments on the lack of robustness of some of those metrics, causing the ranking of the 2018 Data Science Bowl to vary significantly. Then, they stated a recommendation to mitigate the ambiguity of the metric definition by specifying the measure without ambiguity.

The manuscript seems too narrow for a Nat Meth perspective article. The misinterpretations of segmentation metrics are real. But the issue is superficially treated in the manuscript, only one measure

is presented as a use case, and the take-home message is too simplistic. As presented, the message of the manuscript is oriented toward image analysts but far too applicable to life science research.

As real and complex as the treated issue is, it should be approached in a more global and in-depth strategy (e.g., integrating several metrics, standardizing their definition, studying which metric is more appropriate to use in which circumstances, characterizing their robustness for different tasks). A committee representing the community's views could probably be in a good place to take the challenge and collect a relation of best practices to be adopted by the community along the lines of the work performed by MICCAI Challenges Interest Group (Ref. 21 in the manuscript).

Reviewer #2:

Remarks to the Author:

Hirling and colleagues have written a nice review of metrics for image segmentation, and their nomenclatures in the field. I think it is a timely piece, but a bit of clarification in some places would make the review stronger and more readable for someone not directly within the segmentation community:

* A brief comment on the quantification that precision vs recall vs F1 score vs threat score give you. For example that the user cares about high precision when they want to make sure every object is detected, and don't care about false detections, whereas the F1 and threat score penalize false detections, and thus make them more appropriate for cell counting applications.

* It would be good to note that the F1 score and the threat score are deterministically related ($F1 = 2/(1 + 1/\text{threat})$), hence the ~ 1.0 correlation in the figure. It may make sense to show a different metric in the figure to better explore the metric space for comparisons.

* It is very useful to distinguish between the aggregated versus average score across images as the paper does, but I think it's important to discuss why you would use one metric versus the other, depending on the application. For example, in a dataset with many similar images, with some images with many more cells, an aggregated score makes sense because maximizing the detection accuracy across all objects is desired. However, when benchmarking performance on a diverse dataset of images, the average score across images makes sense, because the weighting of the performance on each image is equal. Otherwise, with an aggregated metric, images with more objects will contribute more to the metric than images with fewer objects although the overall goal is to maximize performance across many diverse images.

* It would be useful to mention why many papers in biological image segmentation have chosen metrics like those in Table 1, rather than the computer vision metric AP1 (integral of the precision-recall curve). For example this AP1 metric requires a confidence score for each object, and not all algorithms have such a score. Also, the threshold for this score must be set by the user ultimately to perform segmentation.

Minor:

* Given the lower correlation of the metrics' performances at IoU threshold of 0.5 with IoU threshold of 0.9, can you speculate that metrics at such high thresholds are less useful for determining the quality of the algorithm?

* Why was the digits score chosen to show on the figure? Is it a commonly used score in biological image segmentation, or do you think it should be?

Author Rebuttal to Initial comments

Changes made to the manuscript were highlighted.

Reviewer #1:

Remarks to the Author:

1) The manuscript entitled 'Segmentation metric misinterpretations in bioimage analysis' aims to shed light on some of the issues that appear due to misinterpretation of the definition of the segmentation scores in biomedical images. After a brief introduction and the definition of the main metrics. The manuscript uses the extensively used Average Precision (AP) metric and its variations as a use case. It comments on the lack of robustness of some of those metrics, causing the ranking of the 2018 Data Science Bowl to vary significantly. Then, they stated a recommendation to mitigate the ambiguity of the metric definition by specifying the measure without ambiguity.

We are grateful to the Reviewer for the detailed and constructive review of our work. We hope that our detailed discussions below and the changes in the manuscript will answer all issues raised.

2) The manuscript seems too narrow for a Nat Meth perspective article. The misinterpretations of segmentation metrics are real. But the issue is superficially treated in the manuscript, only one measure is presented as a use case, and the take-home message is too simplistic. As presented, the message of the manuscript is oriented toward image analysts but far too applicable to life science research.

We thank the reviewer for pointing this out. Indeed, we focus on image analysis which we believe is already a large field. Life sciences in general are too broad and we are not aware of many related issues. Discussed equations are applicable to segmentation, to object detection with minor to no modifications. Depending on the analysis task (semantic segmentation, object detection, instance segmentation) different strategies should be used to select the most appropriate measure. E.g. partial overlaps should be handled and a segmentation should be assigned to one of the overlapping objects in the ground truth. We cannot provide similar issues to AP, however, the main message could be the same, that special attention should be made to describe in detail the metric used for the evaluation.

Based on this Reviewer's feedback, we contacted the Editor for advice. She highlighted that the "work is already complementary to the preprint from Maier-Hein et al and can stay focused on image segmentation", thus we would like to keep the focus of the paper in this field.

Nonetheless, to widen the scope, we have extended the manuscript with the analysis of submissions from two other challenges to show that our points stand for different use cases as well.

3) As real and complex as the treated issue is, it should be approached in a more global and in-depth strategy (e.g., integrating several metrics, standardizing their definition, studying which metric is more appropriate to use in which circumstances, characterizing their

robustness for different tasks). A committee representing the community's views could probably be in a good place to take the challenge and collect a relation of best practices to be adopted by the community along the lines of the work performed by MICCAI Challenges Interest Group (Ref. 21 in the manuscript).

We agree with this comment and partially this was our way of thinking when proposing the current paper. The preprint from Maier-Hein et al comprehensively discusses the issues and we made an effort to emphasize their paper in our manuscript. In addition we included the opinion (and as new authors) of the organizers of two influential challenges in the field (MIDOG and Sartorius).

Our plan is to indeed propose the metrics misinterpretation issue for roundtable discussion on forums such as the mentioned MICCAI and others like the Napari community, or the AI4Life EU bioimaging initiatives and others. This manuscript may be an impactful start to kick off such discussion.

Reviewer #2:

Remarks to the Author:

Hirling and colleagues have written a nice review of metrics for image segmentation, and their nomenclatures in the field. I think it is a timely piece, but a bit of clarification in some places would make the review stronger and more readable for someone not directly within the segmentation community:

1) A brief comment on the quantification that precision vs recall vs F1 score vs threat score give you. For example that the user cares about high precision when they want to make sure every object is detected, and don't care about false detections, whereas the F1 and threat score penalize false detections, and thus make them more appropriate for cell counting applications.

We thank the reviewer for pointing this out and agree with the suggestion. Thus, we included further discussion related to the metrics.

2) It would be good to note that the F1 score and the threat score are deterministically related ($F1 = 2/(1 + 1/\text{threat})$), hence the ~ 1.0 correlation in the figure. It may make sense to show a different metric in the figure to better explore the metric space for comparisons.

We thank the reviewer for highlighting the relationship between the F1 and threat score metrics. We have now added a note on it in the manuscript. Panels b and c in Fig. 1. are meant to show metrics with high and low correlation values. We believe that panel b is a great example because in theory, the correlation value should be 1, but in practice, it is slightly different. In this case, this is mainly due to two things: First, the aggregation does not keep this connection of the metrics. Second, sorting algorithms differ in the way they order elements with equal values. We think these observations further justify our main message that small differences in metric interpretations can lead to different leaderboard results.

3) It is very useful to distinguish between the aggregated versus average score across images as the paper does, but I think it's important to discuss why you would use one metric versus the other, depending on the application. For example, in a dataset with many similar images, with some images with many more cells, an aggregated score makes sense

because maximizing the detection accuracy across all objects is desired. However, when benchmarking performance on a diverse dataset of images, the average score across images makes sense, because the weighting of the performance on each image is equal. Otherwise, with an aggregated metric, images with more objects will contribute more to the metric than images with fewer objects although the overall goal is to maximize performance across many diverse images.

We appreciate the suggestion and agree that this aspect should be discussed more thoroughly. Now we have included a more in-depth description of the aggregated vs average score use cases.

4) It would be useful to mention why many papers in biological image segmentation have chosen metrics like those in Table 1, rather than the computer vision metric AP1 (integral of the precision-recall curve). For example this AP1 metric requires a confidence score for each object, and not all algorithms have such a score. Also, the threshold for this score must be set by the user ultimately to perform segmentation.

Thank you for the suggestion, we agree with this. Now we included remarks related to why biological image segmentation uses different metrics.

Minor:

5) Given the lower correlation of the metrics' performances at IoU threshold of 0.5 with IoU threshold of 0.9, can you speculate that metrics at such high thresholds are less useful for determining the quality of the algorithm?

We thank the reviewer for the interesting question. What we can say is that metrics at such high thresholds do not serve the same purpose as those at lower thresholds. In case of segmentation, an IoU threshold of 0.5 essentially means that an object has been found and the quality of the segmentation does not matter as much. On the other hand, if using a threshold of 0.9, pixel-wise segmentation accuracy becomes much more important. With that being said, we think that evaluating metrics across a wide range of IoU thresholds is useful because this way, we can incorporate all of these aspects into one metric.

6) Why was the digits score chosen to show on the figure? Is it a commonly used score in biological image segmentation, or do you think it should be?

The digits score was chosen simply because it is an alternative definition to the mAP. We wanted to compare as many interpretations as possible to show what consequences it had if one chose one over the other.

Decision Letter, first revision:

Dear Peter,

Thank you for submitting your manuscript entitled "Segmentation Metric Misinterpretations in Bioimage Analysis". We think that the revised version should address the referee concerns, and appreciate the additional analyses of other competitions that were carried out.

We discussed your paper at length as a team and ultimately decided that given how much data reanalysis is done that the paper is actually now best suited to be in a research format. We therefore ask you to reformat as a Brief Communication (~1800 words, 3 figures or 2000 words, 2 figures) and include a methods section detailing your approaches used.

We therefore invite you to revise your manuscript to update the format before we send it back to review. We think that ultimately this will make the process faster.

Please ensure that the revised version is as concise as possible, and that it conforms to our format requirements (see <http://www.nature.com/nmeth> for our Guide to Authors).

We shall hope to receive your revised version as soon as you are able to complete the suggested revisions. If something similar is published in the interim we will have to consider the impact it has on the novelty of the revised manuscript.

If you anticipate a delay of more than four weeks, please let us know. In this event, we will still be happy to reconsider your paper at a later date so long as nothing similar has been accepted for publication at Nature Methods or published elsewhere. In the event of publication, however, the received date would be that of the revised rather than the original version.

If you are not interested in submitting a revised manuscript in the future please let me know immediately so we can close your file. If you have any questions, please contact me.

Please use the link below when you are prepared to resubmit.

[Redacted]

The above URL links to your confidential home page and associated information about manuscripts you may have submitted, or that you are reviewing for us. If you wish to forward this email to co-authors, please delete the link to your homepage.

Thank you for your interest in Nature Methods.

Sincerely,
Rita

Rita Strack, Ph.D.
Senior Editor
Nature Methods

Decision Letter, second revision:

Dear Peter,

Thank you for submitting your revised manuscript "Segmentation Metric Misinterpretations in Bioimage Analysis" (NMETH-BC50502B). It has now been seen by the original referees and their comments are below. The reviewers find that the paper has improved in revision, and therefore we'll be happy in principle to publish it in Nature Methods, pending minor revisions to satisfy the referees' final requests and to comply with our editorial and formatting guidelines.

In response to referee 2, we ask that you add a discussion of issues regarding over-interpreting metrics at IoU=0.9.

TRANSPARENT PEER REVIEW

Nature Methods offers a transparent peer review option for new original research manuscripts submitted from 17th February 2021. We encourage increased transparency in peer review by publishing the reviewer comments, author rebuttal letters and editorial decision letters if the authors agree. Such peer review material is made available as a supplementary peer review file. Please state in the cover letter 'I wish to participate in transparent peer review' if you want to opt in, or 'I do not wish to participate in transparent peer review' if you don't. Failure to state your preference will result in delays in accepting your manuscript for publication.

ORCID

Sincerely,
Rita

Rita Strack, Ph.D.
Senior Editor
Nature Methods

Reviewer #1 (Remarks to the Author):

A. Summary of the key results

We propose a revised version of the abstract: 'Quantitative evaluation of image segmentation algorithms is essential in bioimage analysis. However, the most common assessment scores are often misunderstood, with multiple definitions sharing the same name. This article emphasizes the ambiguities of evaluation metrics for segmentation algorithms and illustrates how these misinterpretations can affect the rankings in influential competitions. Furthermore, we provide guidelines on facilitating performance comparisons between algorithms that solve the same task using the same dataset.'

B. Originality and significance

The perspective is original, and the work is significant, holding broad interest for the biomedical imaging community.

C Appropriate use of statistics and treatment of uncertainties

To my knowledge, the use of statistics and the treatment of uncertainties are appropriate.

D. Conclusions: Robustness, validity, reliability

The take-home message is clear and valid.

E. Suggested improvements: experiments, data for possible revision

The current version has significantly improved upon the previous submission. The inclusion of data from three different challenges is highly appreciated.

F. References: appropriate credit to previous work

The reference list appears to be appropriate.

G. Clarity and context: lucidity of abstract/summary, appropriateness of abstract, introduction and conclusions

Overall, the manuscript is clear but would benefit from a style revision.

Reviewer #2 (Remarks to the Author):

Thanks very much for addressing all of my comments, except #5 about the IoU=0.9 metric, which the authors thought did not need to be addressed:

I do think that over-interpreting metrics at IoU=0.9 is problematic for benchmarking methods since the human-to-human variability usually results in low IoU=0.9 accuracies; and as the authors have shown, the metrics are not consistent at IoU of 0.9. Another problem is that in many studies recently, many labels are being made in the loop with an algorithm (e.g. TissueNet paper from Van Valen lab used their algorithm as a starting point for labeling cells). Therefore, the best algorithms for capturing the fine details of the cell outlines will likely be ones that best resemble the starting point for the labeling. These algorithms may not be the most general or the best for example at IoU of 0.5 or 0.75.

Final Decision Letter:

Dear Peter,

Thanks very much for sending me your updated file so quickly.

I am pleased to inform you that your Brief Communication, "Segmentation Metric Misinterpretations in Bioimage Analysis", has now been accepted for publication in Nature Methods. Your paper is tentatively scheduled for publication in our September print issue, and will be published online prior to that. The received and accepted dates will be September 26, 2022 and June 6, 2023. This note is intended to let you know what to expect from us over the next month or so, and to let you know where to address any further questions.

Over the next few weeks, your paper will be copyedited to ensure that it conforms to Nature Methods style. Once your paper is typeset, you will receive an email with a link to choose the appropriate publishing options for your paper and our Author Services team will be in touch regarding any additional information that may be required.

Your paper will now be copyedited to ensure that it conforms to Nature Methods style. Once proofs are generated, they will be sent to you electronically and you will be asked to send a corrected version within 24 hours. It is extremely important that you let us know now whether you will be difficult to contact over the next month. If this is the case, we ask that you send us the contact information (email, phone and fax) of someone who will be able to check the proofs and deal with any last-minute problems.

If, when you receive your proof, you cannot meet the deadline, please inform us at rjsproduction@springernature.com immediately.

Once your manuscript is typeset and you have completed the appropriate grant of rights, you will receive a link to your electronic proof via email with a request to make any corrections within 48 hours. If, when you receive your proof, you cannot meet this deadline, please inform us at rjsproduction@springernature.com immediately.

Once your paper has been scheduled for online publication, the Nature press office will be in touch to confirm the details.

Content is published online weekly on Mondays and Thursdays, and the embargo is set at 16:00 London time (GMT)/11:00 am US Eastern time (EST) on the day of publication. If you need to know the exact publication date or when the news embargo will be lifted, please contact our press office after you have submitted your proof corrections. Now is the time to inform your Public Relations or Press Office about your paper, as they might be interested in promoting its publication. This will allow them time to prepare an accurate and satisfactory press release. Include your manuscript tracking number NMETH-BC50502C and the name of the journal, which they will need when they contact our office.

About one week before your paper is published online, we shall be distributing a press release to news organizations worldwide, which may include details of your work. We are happy for your institution or funding agency to prepare its own press release, but it must mention the embargo date and Nature Methods. Our Press Office will contact you closer to the time of publication, but if you or your Press Office have any inquiries in the meantime, please contact press@nature.com.

If you are active on Twitter, please e-mail me your and your coauthors' Twitter handles so that we may tag you when the paper is published.

Please note that *Nature Methods* is a Transformative Journal (TJ). Authors may publish their research with us through the traditional subscription access route or make their paper immediately open access through payment of an article-processing charge (APC). Authors will not be required to make a final decision about access to their article until it has been accepted. [Find out more about Transformative Journals](https://www.springernature.com/gp/open-research/transformative-journals)

To assist our authors in disseminating their research to the broader community, our SharedIt initiative provides you with a unique shareable link that will allow anyone (with or without a subscription) to read the published article. Recipients of the link with a subscription will also be able to download and print the PDF. As soon as your article is published, you will receive an automated email with your shareable link.

Please note that you and your coauthors may order reprints and single copies of the issue containing your article through Springer Nature Limited's reprint website, which is located at <http://www.nature.com/reprints/author-reprints.html>. If there are any questions about reprints please send an email to author-reprints@nature.com and someone will assist you.

Best regards,
Rita

Rita Strack, Ph.D.
Senior Editor
Nature Methods